# Nitrogen Fertilization in a Faba Bean–Wheat Intercropping System Can Alleviate the Autotoxic Effects in Faba Bean

**DOI:** 10.3390/plants12061232

**Published:** 2023-03-08

**Authors:** Zixuan Cen, Yiran Zheng, Yuting Guo, Siyin Yang, Yan Dong

**Affiliations:** College of Resources and Environment, Yunnan Agricultural University, Kunming 650000, China

**Keywords:** faba bean and wheat intercropping, autotoxicity, seed germination, nitrogen fertilization

## Abstract

Continuous cultivation of the faba bean will lead to its autotoxicity. Faba bean–wheat intercropping can effectively alleviate the autotoxicity of the faba bean. In order to investigate the autotoxicity of water extracts of various parts of the faba bean, we prepared water extracts of various parts of the faba bean, such as the roots, stems, leaves, and rhizosphere soil. The results showed various parts of the faba bean significantly inhibited the germination of faba bean seeds. The main autotoxins in these parts were analyzed using HPLC. Six autotoxins, namely, *p*-hydroxybenzoic acid, vanillic acid, salicylic acid, ferulic acid, benzoic acid, and cinnamic acid, were identified. The exogenous addition of these six autotoxins significantly inhibited the germination of faba bean seeds in a concentration-dependent manner. Furthermore, field experiments were conducted to investigate the effects of various levels of nitrogen fertilizer on the autotoxin content and the aboveground dry weight of the faba bean in a faba bean–wheat intercropping system. The application of various levels of nitrogen fertilizer in the faba bean–wheat intercropping system could significantly reduce the content of autotoxins and increase the aboveground dry weight in faba bean, particularly at the N2 level (90 kg/hm^2^). The above results showed that the water extracts of faba bean roots, stems, leaves, and rhizosphere soil inhibited faba bean seed germination. The autotoxicity in faba bean under continuous cropping could be caused by *p*-hydroxybenzoic acid, vanillic acid, salicylic acid, ferulic acid, benzoic acid, and cinnamic acid. The autotoxic effects in the faba bean were effectively mitigated by the application of nitrogen fertilizer in a faba bean–wheat intercropping system.

## 1. Introduction

Faba bean (*Vicia faba* L.) is the main leguminous crop in China. With the increase in the area of specialized, large-scale, and intensive planting, continuous faba bean planting has caused a high incidence of *Fusarium* wilt, growth inhibition, and a sharp decline in yield [1,2]. The causes of such negative effects of continuous cropping are varied, and the underlying mechanism is very complex. Based on the studies conducted in China and abroad, it is generally believed that the possible causes of the negative effects of continuous cropping include the deterioration of physical and chemical properties and the biological environment of soil and autotoxicity. Studies ha e reported that the accumulation of autotoxins is one of the important factors leading to the negative effects of continuous cropping in agricultural and forest ecosystems [3,4,5].

Autotoxicity is allelopathy within a single plant species [6]. Autotoxins are released from various parts of crops to the environment via secretion, volatilization, rainwater leaching, and decaying of plant residues; the process directly or indirectly inhibits the growth of the crop itself or its relatives in continuous planting [7,8]. Studies have reported that autotoxins are ubiquitous in plant tissues, and the types and amounts of autotoxins present in various parts are different; therefore, the extent of the autotoxic effects in various parts is different. Zhou et al. studied the effect of water extracts of various parts of Elsholtzia stauntoni Benth. on its seed germination; the stem, leaf, and inflorescence exhibited significant autotoxicity with a trend of leaf > inflorescence > stem [9]. Wang et al. reported that the water extract of various parts of stubble of oil flax exhibited autotoxicity on its seed germination, with a trend of stubble leaves > stubble stems > stubble roots [10].

To date, various types of autotoxins have been reported, e.g., phenolic acid compounds, water-soluble organic acids, aliphatic aldehydes, lactones, and long-chain fatty acids 8. Among them, phenolic acids are the most studied and the most active autotoxins [11]. In recent years, more than 10 phenolic acid allelopathic substances have been isolated from various tissues and rhizosphere soils of plants under continuous cropping, among which *p*-hydroxybenzoic acid, ferulic acid, and cinnamic acid are considered the main phenolic acid autotoxic substances. Several studies have reported the autotoxic effects of phenolic acids and proved that phenolic acids can inhibit plant growth by inhibiting plant seed germination [12]. Min et al. reported that ferulic acid exhibited a significant inhibitory effect on wolfberry seed germination [13]. Liang et al. reported that various concentrations of benzoic acid, syringic acid, and vanillic acid solutions could significantly inhibit the germination rate, germination potential, and germination index of Pinus koraiensis seeds [14]. However, these studies focused more on cash crops, and studies on food crops and the autotoxic effects of various parts of the faba bean and the types of autotoxic substances are scarce.

Intercropping, as the essence of Chinese traditional agriculture, is an effective measure to alleviate the effect of autotoxicity and continuous cropping [15]. Through hydroponics and pot experiments, Zhao et al. proved that intercropping with *Perilla frutescens* could effectively overcome the autotoxic effects due to continuous cropping of American ginseng (*Panax quinquefolium*) [16]. Wang et al. studied the effect of various planting patterns on oil flax growth and reported that the intercropping of oil flax and millet could weaken the autotoxic effects of water extracts of soil on seed germination and the growth of oil flax seedlings [17]. Crops need soil to provide nutrients. Fertilization can increase soil nutrient levels, promote crop growth, increase yield, and effectively alleviate the damage caused by autotoxicity to crops. Liu found that adding microbial fertilizer to cotton under continuous cropping could counteract or weaken the allelopathic effect of autotoxins on seed germination [18]. Wang reported that autotoxicity was one of the main causes of the negative effects of the continuous cropping of millet, and the increased application of biochar fertilizer could effectively alleviate this problem [19]. As one of the inorganic nutrients necessary for plant growth, nitrogen has a greater impact on crop yield and quality than other inorganic nutrients [20,21]. However, only a few studies have assessed whether nitrogen fertilizer can alleviate crop autotoxicity. Intercropping and fertilization are both effective ways to alleviate the negative effects of continuous cropping; however, recent studies have focused on a single treatment. Very few studies have reported whether the fertilizer application in the intercropping system can alleviate the autotoxicity and whether the reasonable application of nitrogen fertilizer in the intercropping system can alleviate the autotoxicity.

Faba bean–wheat intercropping is often applied in southwest China to control the negative effects of continuous cropping. Previous studies have reported that faba bean–wheat intercropping can alleviate the autotoxicity in the faba bean; however, it is not clear whether the application of nitrogen fertilizer in this system has the same effect [2,22]. The purpose of this study was: (i) to investigate the autotoxicity on seed germination and the types and contents of phenolic acids in the water extracts of various parts of the faba bean plant and rhizosphere soil to reveal the mechanisms underlying the negative effects of the continuous cropping of the faba bean, and (ii) to study the effects of various levels of nitrogen fertilizer on the phenolic acid content and aboveground biomass of faba bean in the faba bean–wheat intercropping system to provide a theoretical basis for the rational application of nitrogen fertilizer in the intercropping system and the alleviation of continuous cropping obstacles.

## 2. Results

### 2.1. Effect of the Water Extract of Faba Bean Roots on Faba Bean Seed Germination

The water extract of faba bean roots significantly inhibited germination and reduced the GE, GR, and GI of faba bean seeds (Figure 1A–C). Compared with CK, 0.05 and 0.1 g/mL water extract of faba bean roots significantly reduced the GE, GR, and GI of faba bean seeds by 20%, 23%, 3.29 and 37%, 70%, and 7.54, respectively.

### 2.2. Effect of Water Extract of Faba Bean Stems on Faba Bean Seed Germination

The water extract of faba bean stems significantly inhibited faba bean seed germination in a concentration-dependent manner (Figure 2). Compared with CK, the GE of faba bean seed was significantly reduced by 10%, 20%, and 37%; GR was significantly reduced by 13%, 47%, and 77%; and GI was significantly reduced by 2.76, 4.99, and 7.66 after treatment with 0.01, 0.05, and 0.1 g/mL water extract of faba bean stems, respectively (Figure 2A–C). 

### 2.3. Effect of Water Extract of Faba Bean Leaves on Seed Germination of Faba Bean

The water extract of faba bean leaves significantly inhibited the seed germination of faba bean in a concentration-dependent manner (Figure 3). Compared with CK, the GE of faba bean seeds was significantly reduced by 17%, 30%, and 40%; GR of faba bean seeds was significantly reduced by 20%, 57%, and 80%; and GI of faba bean seeds was significantly reduced by 2.65, 5.47, and 7.91, respectively, after treatment with 0.01, 0.05, and 0.1 g/mL water extract of faba bean leaves (Figure 3A–C).

### 2.4. Effect of the Water Extract of Faba Bean Rhizosphere Soil on Faba Bean Seed Germination

The water extract of faba bean rhizosphere soil significantly inhibited the seed germination of the faba bean. It can be seen that the water extract of the faba bean rhizosphere soil significantly reduced the GE, GR, and GI of faba bean seeds. Compared with CK, 0.25 and 0.5 g/mL the water extract of faba bean roots significantly reduced the GE, GR, and GI of faba bean seeds by 17%, 20%, 2.11 and 27%, 30%, 3.67, respectively (Figure 4A–C).

### 2.5. Types and Content of Phenolic Acids in the Water Extract of Faba Bean Roots, Stems, Leaves, and Rhizosphere Soil

The phenolic acids with the highest content in the water extract of the roots, stems, leaves, and rhizosphere soil of the faba bean were cinnamic acid (CA) and salicylic acid (SA), followed by *p*-hydroxybenzoic acid (PA), vanillic acid (VA), ferulic acid (FA), and benzoic acid (BA) (Figure 5).

### 2.6. Effects of Exogenous Phenolic Acids on the Germination of Faba Bean Seeds

PA, FA, BA, CA, VA, and SA were exogenously added to faba bean seeds to study their effects on the germination of faba bean seeds. These phenolic acids exhibited significant allelopathic inhibition on the seed germination of the faba bean in a concentration-dependent manner.

The exogenous addition of various concentrations of PA, FA, BA, CA, VA, and SA significantly reduced the GE, GR, and GI of faba bean seeds, and the RI were <0 (Table 1).

Compared with CK, FA at 100, 200, 400, and 800 mg·L^−1^ significantly reduced the GE of faba bean seeds by 10%, 20%, 38%, 42%; significantly reduced the GR of faba bean seeds by 20%, 28%, 32%, 40%, and significantly reduced the GI of faba bean seeds by 0.92, 1.95, 3.64, 4.08, respectively.

Compared with CK, PA, BA, CA, and SA at 50, 100, 200, 400, and 800 mg·L^−1^ significantly reduced the GE of faba bean seeds by 10%, 18%, 24%, 30%, and 42% (PA); 12%, 24%, 34%, 40%, and 44% (BA); 12%, 24%, 30%, 36%, and 40% (CA); 14%, 22%, 30%, 38%, and 44% (SA), significantly reduced the GR of faba bean seeds by 4%, 14%, 22%, 30%, and 34% (PA); 10%, 18%, 26%, 34%, and 74% (BA); 4%, 10%, 22%, 74%, and 90% (CA); 14%, 34%, 64%, 82%, and 86% (SA), and significantly reduced the GI of faba bean seeds by 0.84, 1.18, 1.67, 2.35, and 3.53 (PA); 1.07, 1.84, 3.19, 4.03, and 5.97 (BA); 0.99, 1.13, 2.58, 5.72, and 6.67 (CA); 1.12, 2.06, 4.50, 6.19, and 6.76 (SA), respectively.

Compared with CK, VA at 50, 100, 200, 400, and 800 mg·L^−1^ significantly reduced the GE of faba bean seeds by 10%, 14%, 22%, 34%, and 36%, respectively. VA at 200, 400, and 800 mg·L^−1^ significantly reduced the GR of faba bean seeds by 16%, 32%, and 46% and GI by 1.57, 3.34, and 4.60, respectively.

### 2.7. Effects of Nitrogen Application and Intercropping on Phenolic Acid Content in the Roots, Stems, Leaves, and Rhizosphere Soil of Faba Bean

Nitrogen application and intercropping significantly reduced the content of phenolic acids in the roots, stems, leaves, and rhizosphere soil of the faba bean (Table 2, Table 3, Table 4 and Table 5).

Compared with N0, N1–N3 significantly reduced the content of PA, VA, FA, BA, SA, and CA in the roots of faba bean by 3–28%, 5–30%, 26–59%, 8–51%, 5–32%, and 15–27%, respectively, under monocropping and by 10–44%, 20–48%, 25–50%, 20–64%, 11–56%, and 16–47%, respectively under intercropping.

Compared with N0, N1–N3 significantly reduced the content of PA, VA, FA, BA, SA, and CA in faba bean stems by 7–24%, 8–28%, 29–48%, 26–51%, 7–14%, and 9–13%, respectively, under monocropping and by 4–47%, 24–49%, 34–60%, 34–62%, 13–19%, and 17–40%, respectively, under intercropping (Table 3).

Compared with N0, N1–N3 significantly reduced the content of PA, VA, FA, BA, SA, and CA in faba bean leaves by 9–17%, 15–47%, 37–53%, 35–47%, 7–12%, and 7–15%, respectively, under monocropping and by 13–32%, 24–64%, 37–72%, 34–66%, 12–26%, and 10–27%, respectively, under intercropping (Table 4).

Compared with N0, N1–N3 significantly reduced the content of PA, VA, FA, BA, SA, and CA in the rhizosphere soil of faba bean by 9–23%, 8–31%, 23–50%, 18–57%, 6–22%, and 18–32%, respectively, under monocropping and by 3–41%, 5–42%, 35–67%, 29–55%, 4–33%, and 9–38%, respectively, under intercropping (Table 5).

Compared with monocropping, intercropping significantly reduced the content of PA, VA, FA, BA, SA, and CA at N0–N3 levels by 12–31%, 18–38%, 24–38%, 17–38%, 10–42%, and 11–35%, respectively, in faba bean roots; by 14–40%, 24–46%, 12–35%, 18–41%, 17–23%, and 18–44%, respectively, in faba bean stems; by 29–42%, 15–42%, 22–56%, 19–49%, 16–29%, and 16–27%, respectively, in faba bean leaves; and by 12–37%, 25–39%, 16–45%, 22–33%, 11–24%, and 9–25%, respectively, in rhizosphere soil of faba bean (Table 2, Table 3, Table 4 and Table 5).

Therefore, nitrogen application and intercropping could significantly reduce the content of the six phenolic acids in faba bean roots, stems, leaves, rhizosphere soil of the faba bean, which reached the lowest level at N2 (Table 2, Table 3, Table 4 and Table 5).

### 2.8. Effects of Nitrogen Application and Intercropping on the Dry Weight of the Faba Bean

Nitrogen application and intercropping significantly increased the aboveground dry weight of the faba bean during the different growth stages of the faba bean (Table 6).

Under monocropping, compared with N0, the aboveground dry weight of the faba bean significantly increased under N2 and N3 by 35% and 32%, respectively, at the branching stage, by 58%, 95%, and 85%, respectively, at N1–N3 at the flowering stage, and by 55%, 96%, and 80%, respectively, at N1–N3 at the maturity stage.

Under intercropping, compared with N0, the aboveground dry weight of the faba bean significantly increased under N1–N3 by 10%, 44%, and 37%, respectively, at the branching stage, by 38%, 49%, and 29%, respectively at the flowering stage, and by 43%, 61%, and 38%, respectively, at the maturity stage.

At different growth stages of the faba bean, intercropping can significantly increase the dry weight of the faba bean under N2 level compared with monocropping. At the branching stage, under N2 treatment, intercropping significantly increased the aboveground dry weight of the faba bean by 7% compared with monocropping; at the flowering stage, under N2 treatment, intercropping significantly increased the aboveground dry weight of the faba bean by 18% compared with monocropping; at the maturity stage, under N2 treatment, intercropping significantly increased the aboveground dry weight of the faba bean by 26% compared with monocropping.

Therefore, the dry weight of aboveground part of the faba bean was significantly increased after nitrogen fertilizer application at various stages of the disease, reaching the maximum level at N2.

## 3. Discussion

Faba beans have been cultivated for more than 2000 years in China, which has the largest cultivation area and largest proportion of total output of faba beans worldwide. Due to the expanding market demand and other factors, continuous cropping of the faba bean has increased, causing serious negative effects such as plant production inhibition, decline in yield, and high incidence of diseases [2]. In recent years, many studies have confirmed the serious effects of the continuous cropping of melon [23], Angelica sinensis [24], and strawberry [25].

The negative effects of continuous cropping are attributed to various complex factors; however, studies in China and abroad have generally attributed them to autotoxicity [3,4,5]. Plants can release autotoxins to the environment in various ways through various organs. Autotoxins mainly include secretions by plant roots, leachates, volatile compounds from plant stems and leaves, and substances produced by the decomposition of plant residues. With time, the autotoxin-released plant organs are continuously accumulated, eventually leading to autotoxic effects and inhibiting the normal growth and development of plants [26]. Autotoxins are secondary metabolites of plants, mainly including water-soluble phenolic acids, organic acid, alcohols, and aldehydes. Currently, phenolic acids are the most widely studied autotoxic substances.

Compared with organic solvent extracts, water extracts of plants can directly reflect the autotoxicity of plants [27]. One of the main methods to study autotoxicity is to examine the effect of autotoxins on seed germination. Seed germination is an important prerequisite for plant growth. GR is the most direct indicator of seed germination ability [28]. GE can reflect the growth ability of the seeds. The higher the GE is, the more resistant the seed is to adversity, and it is an important indicator to guide crop growth [29]. GI can reflect seed vigor to a certain extent [30]. Seed germination regulates the yield and productivity of plants, which is of great significance to the growth, development, and yield of plants in the later period. The research results are representative [12]. Huang et al. studied the autotoxicity of the water extracts of the leaves, stems, roots, and rhizosphere soil of the peanut. The leaf extract of the peanut exhibited the strongest inhibitory effect on seed germination, followed by the stem and root extracts. In addition, four autotoxic phenolic acids (*p*-hydroxybenzoic acid, vanillic acid, coumaric acid, and coumarin) were detected in the rhizosphere soil [31]. Wang et al. studied the autotoxic effect of water extracts of various parts of oil flax at various concentrations (1, 2.5, 5, 7.5, 10, 15, 25, 50, 75, and 100 mg·mL^−1^) on the seed germination and observed that the autotoxic effect of various parts was different, exhibiting a trend of leaf > stem > root. Overall, 29, 34, and 16 autotoxic substances were found in the leaves, stems, and roots of oil flax, respectively 10. Li reported that the extracts of *p*-hydroxybenzoic acidsoybean root, stem, leaf, and rhizosphere soil at 0.01, 0.04, and 0.16 g·mL^−1^ exhibited various degrees of inhibition on the seed germination in a concentration-dependent manner, exhibiting a trend of leaf > soil > root > stem [32]. Phthalate, cinnamic acid, and were simultaneously detected in various extracts of the adzuki bean, and the substances with a high content were all proved to exhibit autotoxicity via biological verification.

These results are consistent with our study. In our study, the water extract of roots, stems, and leaves at 0.01, 0.05, and 0.1 g/mL and those of rhizosphere soil at 0.025, 0.25, and 0.5 g/mL significantly inhibited the GE, GR, and GI of faba bean seeds in a concentration-dependent manner, exhibiting a trend of leaf > stem > root > rhizosphere soil. We identified the types of phenolic acids in the water extract of various parts of the faba bean, and *p*-hydroxybenzoic acid, vanillic acid, salicylic acid, ferulic acid, benzoic acid, and cinnamic acid exhibited high contents. Overall, the contents of cinnamic acid and salicylic acid were the highest (the highest in the water extract of leaves). Various concentrations (50, 100, 200, 400, and 800 mg·L^−1^) of these phenolic acids were exogenously added to faba bean seeds, after which a significant inhibitory effect was observed on the GE, GR, and GI of the faba bean.

The RI values of faba bean seeds germinated after treatment with the six phenolic acids were <0, and the RI values decreased with the increase in the concentration of phenolic acids [33]. *P*-hydroxybenzoic acid, vanillic acid, salicylic acid, ferulic acid, benzoic acid, and cinnamic acid exhibited significant autotoxic effect on faba bean seeds in a concentration-dependent manner. Therefore, the water extract of root, stem, leaves, and rhizosphere soil of the faba bean exhibited a significant autotoxic effect on the germination of faba bean seeds, and *p*-hydroxybenzoic acid, vanillic acid, salicylic acid, ferulic acid, benzoic acid, and cinnamic acid may have been the autotoxins.

Previous studies have reported that autotoxins can be produced in various plant tissues, including leaves, stems, roots, and seeds; however, their levels vary in various tissues [34]. Leaves are the most stable source of autotoxic substances, with a strong autotoxic effect. However, only a few autotoxins are present in roots, and the autotoxic effect is also small [35]. This may be the reason why the autotoxicity of water extracts from various plant parts is different. In addition, phenolic acids such as cinnamic acid have been proved to be autotoxins in many planting systems [36,37]. The inhibition of seed germination by autotoxins could be via inhibiting seed germination by inhibiting key enzymes required for seed germination [38], inhibiting seed germination by inhibiting the growth of embryo axis and radicle [39], or affecting seed germination by preventing cell division and inhibiting cell elongation [40]. In summary, the intensity of the autotoxicity of various plant parts is different, which may be due to the different levels of autotoxins in various parts. The content of autotoxins is more in plant leaves; therefore, the autotoxic effect of leaves is stronger. Therefore, in agricultural practice, it is necessary to clean up dead branches and fallen leaves in a timely manner, reduce the accumulation of autotoxins in the fallen leaves in the soil, and prepare and loosen the soil to prevent excessive accumulation of autotoxins from the previous crop, which will affect the cultivation of the next crop.

As the essence of Chinese traditional agriculture, intercropping is an effective measure to alleviate the negative effects of continuous cropping by combining land use and land maintenance in agricultural production [41]. Reasonable intercropping can reduce the autotoxic effect and negative effect of continuous cropping [42,43]. Through the water culture and pot experiment, Zhao et al. proved that American ginseng and P. frutescens intercropping could effectively overcome the autotoxic effect of the continuous cropping of American ginseng [16]. Wang et al. reported that the intercropping of oil flax and wheat can reduce the autotoxicity in oil flax [17]. These results are consistent with our study. Our study demonstrated that faba bean–wheat intercropping can significantly reduce the content of *p*-hydroxybenzoic acid, vanillic acid, salicylic acid, ferulic acid, benzoic acid, and cinnamic acid in the root, stem, leaf, and rhizosphere soil of the faba bean. This could be the reason why faba bean–wheat intercropping could alleviate the autotoxicity in the faba bean to a certain extent. In addition, the decrease in phenolic acid content under intercropping may be related to the change in the soil microbial community and the reduction in the abundance of soil-borne pathogens [44]. The composition of the soil microbial community is strongly affected by farming methods [45]. Studies have reported that mixed exoplasts of various plants can change the abundance and diversity of bacterial and fungal communities in soil microbes. Therefore, intercropping may reduce the abundance of pathogenic bacteria in the rhizosphere soil and increase the abundance of beneficial microbes that can degrade phenolic acids, thus reducing the content of phenolic acids [22].

Crops need soil to obtain nutrients for growth and the nutrient supply will affect its own toxicity; therefore, fertilization can effectively alleviate the autotoxicity to a certain extent [46]. However, unbalanced fertilization cannot alone alleviate the autotoxicity but may increase the accumulation of autotoxins in the soil, making the situation worse [47]. It has become an inevitable trend to mitigate the negative effects of continuous cropping and improve the crop yield through the rational application of nitrogen fertilizer [48]. However, excessive nitrogen fertilizer input hinders the development of the plant root system, resulting in an overgrowth of crops and reduced resistance [32]. Our study demonstrated that nitrogen fertilizer application in the faba bean–wheat intercropping system could significantly reduce the content of *p*-hydroxybenzoic acid, vanillic acid, salicylic acid, ferulic acid, benzoic acid, and cinnamic acid in the root, stem, leaf, and rhizosphere soil of the faba bean. The content of autotoxins was the lowest at N2. This could be because the reasonable application of nitrogen fertilizer provides a large amount of substrate for an enzymatic reaction in the soil and promotes the reaction of polyphenol oxidase in the soil. Polyphenol oxidase is a compound enzyme released by plant roots and via soil microbial activities and decomposition of animal and plant residues, which can degrade phenolic acids in the soil and slow down the autotoxicity of plants [49]. Furthermore, our study reported that nitrogen fertilization in the faba bean–wheat intercropping system could significantly increase the aboveground biomass of the faba bean, particularly at the N2 level. The intercropping system of legumes and gramineous crops, along with optimum fertilization can increase the biomass of legumes because rational fertilization makes gramineous plants absorb and use more soil nitrogen, reducing the nitrogen concentration in the soil. This enables gramineous crops to obtain sufficient nitrogen nutrition, which has a significant role in increasing production, and the reduction in soil nitrogen concentration promotes the nodulation and nitrogen fixation by leguminous crops; therefore, both crops can achieve a high yield [50]. Therefore, in future agricultural production, the optimum application of nitrogen fertilizer in a reasonable intercropping mode can be a more cost-effective and environmentally friendly measure to reduce the negative impact of continuous cropping and improve crop yield.

## 4. Materials and Methods

### 4.1. Experimental Materials

In this study, faba bean variety 89–147 and wheat variety Yunmai 42 were purchased from Yunnan Academy of Agricultural Sciences. Analytical grade *p*-hydroxybenzoic acid, vanillic acid, salicylic acid, ferulic acid, benzoic acid, and cinnamic acid were purchased from China Pharmaceutical Group Shanghai Medical Equipment Co., Ltd. (Shanghai, China). Phenolic acid standard samples (chromatographically pure) detected by HPLC were purchased from Sigma Company (Ronkonkoma, NY, USA).

### 4.2. Preparation of the Water Extract from Faba Bean Plant and Rhizosphere Soil

At the maturity stage of the faba bean, the whole plants with *Fusarium* wilt disease were collected from the test field. Dust adhered to the plants and roots was cleaned (first washed with tap water and then with deionized water). The roots, stems, and leaves were separated and heated in an oven at 105 °C for 30 min, followed by drying at 65 °C till a constant weight was obtained. Further, the parts were cut into 1-cm-long segments. To 20 g dry samples of roots, stems, and leaves, 200 mL deionized water was added and shaken at constant temperature for 2 h. The samples were subjected to extraction at room temperature for 48 h. The extracts were filtered with three layers of gauze and centrifugated at 4000 r·min^−1^ for 10 min. The supernatant was collected, considered as 0.1 g·mL^−1^ stock solution of extract, and stored at −20 °C.

The rhizosphere soil of the faba bean after 6 years of continuous cropping was collected, naturally air dried, ground, sieved, and soaked in a beaker according in a ratio soil:water = 1:2. The suspension was vigorously shaken and allowed to stand overnight. Further, it was filtered through three layers of gauze and centrifugated for 10 min at 2000 rpm. The supernatant was considered as 0.5 g·mL^−1^ stock solution of rhizosphere soil extract and stored at −20 °C.

### 4.3. Determination of the Biological Activity of Faba Bean Seed Germination

#### 4.3.1. Determination of the Water Extract of Various Parts of the Faba Bean on the Germination of Faba Bean Seeds

Overall, 10 faba bean seeds of the same size were selected and placed in a 15-cm culture dish covered with filter paper. To this, 20 mL water extracts of roots, stems, and leaves of concentrations 0.01, 0.05, and 0.1 g·mL^−1^, each, and those of rhizosphere soil at 0.025, 0.25, and 0.5 g·mL^−1^ were added. An equivalent amount of sterile water was added to the control CK. Each treatment was repeated three times. The plates were incubated in a constant temperature light incubator at 25 °C for 7 days; water extract and sterile water were regularly replaced. The germination was observed every day, and the number of germinated seeds was recorded. When the radicle broke through the seed coat and its length was half of the seed length, the seed was considered germinated. The germination potential was evaluated after 3 days, and the germination rate and germination index were measured 7 days later as per the formulae given below.

#### 4.3.2. Determination of the Effect of Phenolic Acids on the Germination of Faba Bean Seeds

The biological activity of phenolic acids was determined via the exogenous addition method. This experiment was conducted in the plastic shed of Houshan Farm, Yunnan Agricultural University, from October 2015 to March 2016. The concentrations of *p*-hydroxybenzoic acid, vanillic acid, salicylic acid, ferulic acid, benzoic acid, and cinnamic acid (analytical grade) used were 50, 100, 200, 400, and 800 mg·L^−1^. An equivalent amount of sterile water was used as the control CK.

Faba bean seeds of the same size, same plumpness, and complete seed coat were selected, soaked in 10% H_2_O_2_ for 30 min, cleaned with deionized water, and put into a petri dish containing various phenolic acids. A filter paper was placed at the bottom of the petri dish. Each treatment was repeated 5 times with 10 seeds each time. The cotyledons were transplanted into Hoagland nutrient solution for culture. While planting, 4–5 plants were placed in each pot, and the nutrient solution was changed every 2 days. A ventilation pump was used for ventilation during the test. The pH of the nutrient solution was maintained between 6.2 and 6.3. The GE was calculated on the third day, and the GR and GI were calculated 7 days later.

(1)Germination energy (GE)
(1)$$\mathrm{GE}=\frac{\mathrm{Germination}\;\mathrm{number}\;\mathrm{of}\;\mathrm{test}\;\mathrm{seeds}\;\mathrm{within}\;3\;\mathrm{days}}{\mathrm{number}\;\mathrm{of}\;\mathrm{test}\;\mathrm{seeds}}\times 100\%$$(2)Germination rate (GR)
(2)$$\mathrm{GR}=\frac{\mathrm{Germination}\;\mathrm{number}\;\mathrm{of}\;\mathrm{test}\;\mathrm{seeds}\;\mathrm{within}\;7\;\mathrm{days}}{\mathrm{number}\;\mathrm{of}\;\mathrm{test}\;\mathrm{seeds}}\times 100\%$$(3)Germination index (GI)
(3)$$\mathrm{GI}=\sum^{\;}\left(G_{t}/D_{t}\right)$$where G_t_ is the number of germinated seeds on day t, and D_t_ is the number of days of germination.(4)response index (RI)
(4)RI=1−C/TC: CK; T: treatment

### 4.4. Determination of Phenolic Acid Composition and Content

*P*-hydroxybenzoic acid, vanillic acid, salicylic acid, ferulic acid, benzoic acid, and cinnamic acid (HPLC grade) were taken as the standards for HPLC. The HPLC conditions were as follows: chromatographic column: Kinetex column, 2.6 μm, 4.6 × 100 mm, column temperature 30 °C; injection volume, 10 μL; 280 nm DAD detector; flow rate, 0.5 mL·min^−1^; mobile phase: A = methanol (chromatographic grade), B = 0.1% phosphoric acid in water; elution conditions: mobile phase B 80% (0 min) → 5% (15.0 min) → 5% (18.0 min) → 80% (18.5 min) → 0% (20.0 min) → stop (25.0 min) for gradient elution. The type of phenolic acid was determined according to the retention time, and the content of each phenolic acid was calculated using the external standard method.

### 4.5. Field Test

The field test was conducted in Fengcun Village, Eshan County, Yuxi City, Yunnan Province, from October 2017 to May 2018 (24°11′ N, 102°24′ E, elevation 1540 m, monthly average temperature 18.4 °C, and annual rainfall 582.2 mm). The test soil type was paddy soil. Before the field experiment arrangement in October 2017, the soil nutrient content was as follows: organic matter 28.6 g·kg^−1^, total nitrogen 2.6 g·kg^−1^, total phosphorus 0.75 g·kg^−1^, total potassium 18.2 g·kg^−1^, alkali hydrolyzed nitrogen 108 mg·kg^−1^, available phosphorus 34.2 mg·kg^−1^, available potassium 97.4 mg·kg^−1^, and pH 6.7 in the 0–20-cm topsoil.

The field experiment involved a two-factor design (A, B): A was the planting mode: faba bean monocropping and faba bean–wheat intercropping; B refers to four levels of nitrogen application [no nitrogen (N0; 0 kg/hm^2^), low nitrogen (N1; 45 kg/hm^2^), conventional nitrogen (N2; 90 kg/hm^2^), and high nitrogen (N3; 135 kg/hm^2^)]. There were 8 treatments in total, and each treatment was repeated 3 times. There were 24 plots in total, and the area of each experimental cell was 5.4 × 6 = 32.4 m^2^, in a completely random block arrangement.

In the faba bean monocropping system, the plant spacing was 0.1 m, and the row spacing was 0.3 m. In the faba bean–wheat intercropping system, the faba bean plant spacing and row spacing were 0.1 and 0.3 m, respectively. The wheat plant row spacing was 0.2 m, and the faba bean–wheat intercropping system was 0.3 m. Overall, 18 rows of faba bean were planted in the faba bean monocropping plot. In the faba bean–wheat intercropping plot, 2 rows of faba bean and 6 rows of wheat were sown alternately, with a planting ratio of 1:3. In total, 8 rows of faba bean and 18 rows of wheat were planted. When sowing, the faba bean was sown on demand, and wheat was sown in drill.

The N, P, and K fertilizers used during the test were urea (N 46%), ordinary superphosphate (P_2_O_5_ 16%), and potassium sulfate (K_2_O 50%), respectively. The nitrogen application rate to faba bean was 0, 45, 90, and 135 kg/hm^2^ and that to wheat was 135 kg/hm^2^. The nitrogen fertilizer applied to wheat was divided into base fertilizer and topdressing (1/2 for each) twice. The base fertilizer was applied at the time of sowing, and topdressing was performed at the jointing stage of wheat. The amount of phosphate and potassium fertilizers applied to faba bean and wheat was 90 kg/hm^2^, and they were applied once at the time of sowing. No organic fertilizer was applied during crop growth. No pesticides or agronomic measures were used during the growth of the faba bean.

Three faba bean plants were randomly collected from each plot in the field at the branching stage, flowering stage, and mature stage of the faba bean. The whole fresh faba bean plant was washed with sterile water, and further, the stem, leaves, and root were separated. They were heated at 105 °C for 30 min and cooled to 60–70 °C. Further, they were dried till a constant weight was obtained, placed in bags, weighed, and labelled.

### 4.6. Statistical Analysis

All data were plotted using the origin 2018 (OriginLab, Northampton, MA, USA) and Excel 2010. All data were analyzed using SPSS v26.0 (IBM, Inc., Armonk, NY, USA). The single factor test was performed using the single factor analysis of variance (ANOVA) to evaluate the difference between the treatments of the same substance, and further, LSD test was performed at the 5% probability level. All data were expressed as mean ± standard error. The two factor experiment involved a two factor ANOVA to evaluate the significant differences between the treatments, and further, Tukey’s test was conducted at the 5% probability level. All data were expressed as mean ± standard error.

## 5. Conclusions

The water extract of faba bean roots, stems, leaves, and rhizosphere soil exhibited autotoxic effects on the seed germination, and the relationship of its autotoxicity was leaf > stem > root > rhizosphere soil. Furthermore, *p*-hydroxybenzoic acid, vanillic acid, salicylic acid, ferulic acid, benzoic acid, and cinnamic acid may have been the autotoxic substances underlying the negative effects of the continuous cropping of the faba bean. Reasonable application of nitrogen fertilizer in a faba bean–wheat intercropping system can effectively alleviate the autotoxicity of the faba bean.

## Figures and Tables

**Figure 1 plants-12-01232-f001:**
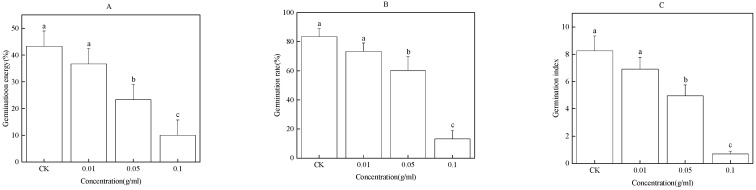
Effect of water extract of faba bean roots on germination energy (GE) (**A**), germination rate (GR) (**B**), and germination index (GI) (**C**) of faba bean seeds. Data are presented as the mean ± standard error of three biological replicates. Various letters for each index indicate significant differences at *p* < 0.05.

**Figure 2 plants-12-01232-f002:**
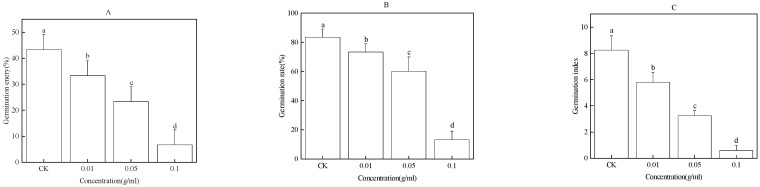
Effect of the water extract of faba bean stems on GE (**A**), GR (**B**), and GI (**C**) of faba bean seeds. Data are presented as the mean ± standard error of three biological replicates. Various letters for each index indicate significant differences at *p* < 0.05.

**Figure 3 plants-12-01232-f003:**
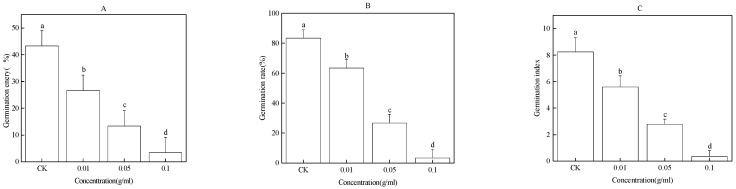
Effect of water extract of faba bean leaves on GE (**A**), GR (**B**), and GI (**C**) of faba bean seeds. Data are presented as the mean ± standard error of three biological replicates. Various letters for each index indicate significant differences at *p* < 0.05.

**Figure 4 plants-12-01232-f004:**
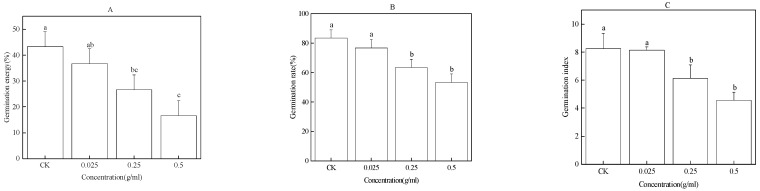
Effect of the water extract of faba bean rhizosphere soil on GE (**A**), GR (**B**), and GI (**C**) of faba bean seeds. Data are presented as the mean ± standard error of three biological replicates. Various letters for each index indicate significant differences at *p* < 0.05.

**Figure 5 plants-12-01232-f005:**
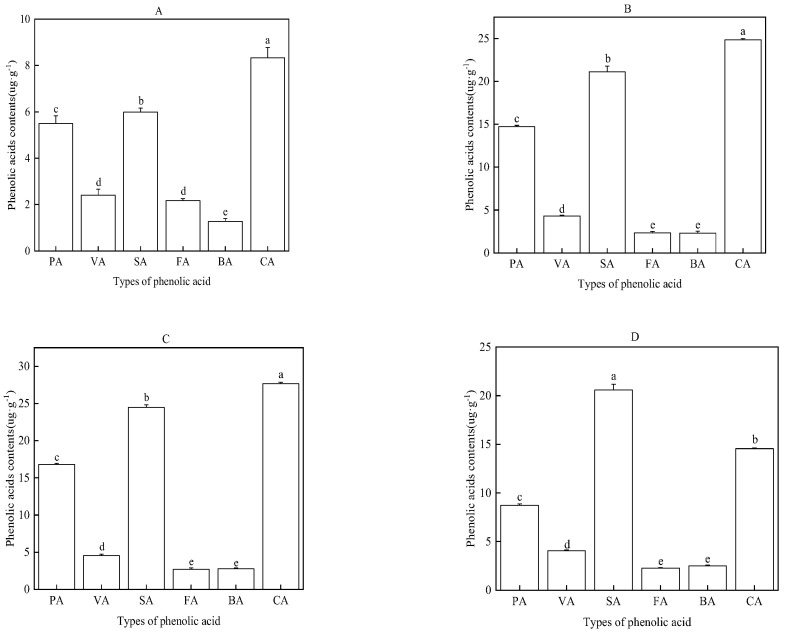
Types and contents of phenolic acids in the water extract of faba bean roots (**A**), stems (**B**), leaves (**C**), and rhizosphere soil (**D**). Data are expressed as the mean ± standard error of three biological replicates. Various letters for each index indicate significant differences at *p* < 0.05.

**Table 1 plants-12-01232-t001:** Effect of the exogenous addition of various phenolic acids on the germination of faba bean seeds.

Types of Phenolic Acids	Concentration (mg·L^−1^)	Germination Energy (%)	RI	Germination Rate (%)	RI	Germination Index (%)	RI
	CK	46.00 ± 5.48 a	−	96.00 ± 5.48 a	−	7.32 ± 0.67 a	−
PA	50	36.00 ± 5.48 b	−0.22	92.00 ± 8.37 a	−0.04	6.48 ± 0.67 b	−0.11
	100	28.00 ± 4.47 c	−0.39	82.00 ± 8.37 b	−0.15	6.14 ± 0.24 bc	−0.16
	200	22.00 ± 4.47 cd	−0.52	74.00 ± 5.48 bc	−0.23	5.65 ± 0.42 c	−0.23
	400	16.00 ± 5.48 d	−0.65	66.00 ± 5.48 cd	−0.31	4.98 ± 0.56 d	−0.32
	800	4.00 ± 5.48 e	−0.91	62.00 ± 4.47 d	−0.35	3.80 ± 0.15 e	−0.48
	CK	46.00 ± 5.48 a	−	96.00 ± 5.48 a	−	7.32 ± 0.67 a	−
FA	50	42.00 ± 4.47 ab	−0.09	90.00 ± 7.07 a	−0.06	6.74 ± 0.54 ab	−0.08
	100	36.00 ± 5.48 b	−0.22	76.00 ± 5.48 b	−0.21	6.41 ± 0.64 b	−0.13
	200	26.00 ± 5.48 c	−0.43	68.00 ± 4.47 c	−0.29	5.37 ± 0.15 c	−0.27
	400	8.00 ± 4.47 d	−0.83	64.00 ± 5.48 c	−0.33	3.69 ± 0.22 d	−0.50
	800	4.00 ± 5.48 d	−0.91	56.00 ± 5.48 d	−0.42	3.25 ± 0.62 d	−0.56
	CK	46.00 ± 5.48 a	−	96.00 ± 5.48 a	−	7.32 ± 0.67 a	−
BA	50	34.00 ± 5.48 b	−0.26	86.00 ± 5.48 b	−0.10	6.25 ± 0.62 b	−0.15
	100	22.00 ± 4.47 c	−0.52	78.00 ± 8.37 c	−0.19	5.49 ± 0.36 c	−0.25
	200	12.00 ± 4.47 d	−0.74	70.00 ± 7.07 d	−0.27	4.13 ± 0.22 d	−0.44
	400	6.00 ± 5.48 de	−0.87	62.00 ± 4.47 e	−0.35	3.29 ± 0.36 e	−0.55
	800	2.00 ± 4.47 e	−0.96	22.00 ± 4.47 f	−0.77	1.35 ± 0.36 f	−0.82
	CK	46.00 ± 5.48 a	−	96.00 ± 5.48 a	−	7.32 ± 0.67 a	−
CA	50	34.00 ± 5.48 b	−0.26	92.00 ± 4.47 ab	−0.04	6.34 ± 0.66 b	−0.13
	100	22.00 ± 4.47 c	−0.52	86.00 ± 8.94 b	−0.10	6.20 ± 0.51 b	−0.15
	200	16.00 ± 5.48 cd	−0.65	74.00 ± 5.48 c	−0.23	4.74 ± 0.50 c	−0.35
	400	10.00 ± 0.00 de	−0.78	22.00 ± 4.47 d	−0.77	1.60 ± 0.56 d	−0.78
	800	6.00 ± 5.48 e	−0.87	6.00 ± 5.48 e	−0.94	0.66 ± 0.60e	−0.91
	CK	46.00 ± 5.48 a	−	96.00 ± 5.48 a	−	7.32 ± 0.67 a	−
VA	50	36.00 ± 5.48 b	−0.22	94.00 ± 5.48 a	−0.02	6.75 ± 0.92 a	−0.08
	100	32.00 ± 4.47 b	−0.30	92.00 ± 4.47 a	−0.04	6.60 ± 0.56 a	−0.10
	200	24.00 ± 5.48 c	−0.48	80.00 ± 7.07 b	−0.17	5.75 ± 0.47 b	−0.21
	400	12.00 ± 4.47 d	−0.74	64.00 ± 5.48 c	−0.33	3.99 ± 0.38 c	−0.46
	800	10.00 ± 0.00 d	−0.78	50.00 ± 0.00 d	−0.48	2.72 ± 0.49 d	−0.63
	CK	46.00 ± 5.48 a	−	96.00 ± 5.48 a	−	7.32 ± 0.67 a	−
SA	50	32.00 ± 4.47 b	−0.30	82.00 ± 4.47 b	−0.15	6.21 ± 0.46 b	−0.15
	100	24.00 ± 5.48 c	−0.48	62.00 ± 4.47 c	−0.35	5.26 ± 0.56 c	−0.28
	200	16.00 ± 5.48 d	−0.65	32.00 ± 4.47 d	−0.67	2.82 ± 0.39 d	−0.61
	400	8.00 ± 4.47 e	−0.83	14.00 ± 5.48 e	−0.85	1.14 ± 0.60 e	−0.85
	800	2.00 ± 4.47 e	−0.96	10.00 ± 0.00 e	−0.90	0.56 ± 0.38 e	−0.92

Data are presented as the mean ± standard error of five biological replicates. Various letters for each index indicate significant differences at *p* < 0.05.

**Table 2 plants-12-01232-t002:** Effects of nitrogen application and intercropping on phenolic acid content in faba bean roots.

Site of Measurement	N Level (N)	Planting Pattern (P)	Content of Phenolic Acid (μg/g)					
			PA	VA	FA	BA	SA	CA
Roots	N0	M	9.48 ± 0.30 a	5.42 ± 0.18 a	3.86 ± 0.12 a	5.48 ± 0.36 a	9.01 ± 0.29 a	11.31 ± 0.67 a
		I	8.37 ± 0.11 A*	4.42 ± 0.35 A*	2.39 ± 0.17 A*	4.56 ± 0.18 A*	8.13 ± 0.09 A*	10.08 ± 0.35 A*
	N1	M	9.19 ± 0.16 a	5.16 ± 0.12 a	2.54 ± 0.09 c	5.01 ± 0.37 a	8.52 ± 0.34 a	9.58 ± 0.36 b
		I	7.56 ± 0.15 B*	3.39 ± 0.19 B*	1.75 ± 0.13 B*	3.66 ± 0.21 B*	7.28 ± 0.06 B*	8.42 ± 0.16 B*
	N2	M	6.83 ± 0.33 c	3.77 ± 0.07 c	1.59 ± 0.26 d	2.68 ± 0.16 c	6.16 ± 0.18 c	8.27 ± 0.38 c
		I	4.68 ± 0.31 C*	2.33 ± 0.13 C*	1.21 ± 0.09 C*	1.66 ± 0.23 D*	3.58 ± 0.13 D*	5.38 ± 0.26 D*
	N3	M	8.75 ± 0.11 b	4.66 ± 0.17 b	2.85 ± 0.09 b	3.64 ± 0.21 b	7.42 ± 0.39 b	9.65 ± 0.40 b
		I	7.29 ± 0.20 B*	3.54 ± 0.27 B*	1.80 ± 0.15 B*	2.31 ± 0.26 C*	6.38 ± 0.44 C*	7.78 ± 0.21 C*
N level (N)			**	**	**	**	**	**
Planting pattern (P)			**	**	**	**	**	**
N × P			**	*	**	ns	**	*

The same upper or lowercase letters in the same column indicate significant differences among various nitrogen application levels under intercropping or monocropping planting patterns, respectively, at *p* < 0.05. * indicates that under the same nitrogen fertilizer application rate, the difference between monocropping and intercropping is significant (*p* < 0.05), and ** is extremely significant (*p* < 0.01), and ns is no significant difference. Data are presented as the mean ± standard error of three biological replicates.

**Table 3 plants-12-01232-t003:** Effects of nitrogen application and intercropping on phenolic acid content in faba bean stems.

Site of Measurement	N Level (N)	Planting Pattern (P)	Content of Phenolic Acid (μg/g)					
			PA	VA	FA	BA	SA	CA
Stems	N0	M	19.50 ± 0.28 a	6.02 ± 0.18 a	4.50 ± 0.26 a	4.74 ± 0.18 a	24.70 ± 0.14 a	28.48 ± 0.32 a
		I	16.32 ± 0.06 A*	4.55 ± 0.19 A*	3.94 ± 0.18 A*	3.88 ± 0.29 A*	20.37 ± 0.45 A*	23.29 ± 0.26 A*
	N1	M	16.53 ± 0.25 c	5.57 ± 0.34 b	3.42 ± 0.14 b	3.45 ± 0.24 b	22.63 ± 0.20 b	26.02 ± 0.82 b
		I	13.45 ± 0.29 C*	3.47 ± 0.19 B*	2.61 ± 0.25 B*	2.55 ± 0.11 B*	17.60 ± 0.09 B*	19.29 ± 0.35 B*
	N2	M	14.71 ± 0.16 d	4.30 ± 0.11 d	2.35 ± 0.16 c	2.32 ± 0.23 c	21.11 ± 0.68 c	24.83 ± 0.15 c
		I	8.70 ± 0.19 D*	2.33 ± 0.13 C*	1.54 ± 0.12 C*	1.49 ± 0.27 D*	16.47 ± 0.25 C*	13.92 ± 0.33 D*
	N3	M	18.12 ± 0.19 b	4.80 ± 0.16 c	3.65 ± 0.19 b	3.51 ± 0.28 b	22.99 ± 0.24 b	25.31 ± 0.57 bc
		I	15.60 ± 0.24 B*	3.31 ± 0.17 B*	2.38 ± 0.16 B*	2.07 ± 0.19 C*	17.65 ± 0.11 B*	18.70 ± 0.16 C*
N level (N)			**	**	**	**	**	**
Planting pattern (P)			**	**	**	**	**	**
N × P			**	*	*	ns	ns	**

The same upper or lowercase letters in the same column indicate significant differences among various nitrogen application levels under intercropping or monocropping planting patterns, respectively, at *p* < 0.05. * indicates that under the same nitrogen fertilizer application rate, the difference between monocropping and intercropping is significant (*p* < 0.05), and ** is extremely significant (*p* < 0.01), and ns is no significant difference. Data are presented as the mean ± standard error of three biological replicates.

**Table 4 plants-12-01232-t004:** Effects of nitrogen application and intercropping on phenolic acid content in faba bean leaves.

Site of Measurement	N Level	Planting Pattern	Content of Phenolic Acid (μg/g)					
			PA	VA	FA	BA	SA	CA
Leaves	N0	M	20.14 ± 0.53 a	8.51 ± 0.26 a	5.79 ± 0.20 a	5.28 ± 0.13 a	27.93 ± 0.33 a	32.69 ± 0.20 a
		I	14.25 ± 0.16 A*	7.26 ± 0.11 A*	4.37 ± 0.26 A*	4.24 ± 0.10 A*	23.35 ± 0.34 A*	27.60 ± 0.43 A*
	N1	M	18.27 ± 0.23 b	7.22 ± 0.09 b	3.63 ± 0.32 b	3.43 ± 0.15 b	25.94 ± 0.35 b	28.74 ± 0.3 c
		I	11.86 ± 0.31 B*	5.48 ± 0.10 B*	2.62 ± 0.14 B*	2.79 ± 0.20 B*	20.64 ± 0.22 B*	22.06 ± 0.35 C*
	N2	M	16.81 ± 0.15 d	4.55 ± 0.18 d	2.72 ± 0.19 c	2.78 ± 0.11 d	24.48 ± 0.34 c	27.66 ± 0.21 d
		I	9.65 ± 0.36 C*	2.65 ± 0.31 D*	1.21 ± 0.09 C*	1.42 ± 0.07 D*	17.28 ± 0.24 D*	20.28 ± 0.85 D*
	N3	M	17.45 ± 0.23 c	6.50 ± 0.12 c	3.47 ± 0.13 b	3.5 ± 0.11 c	25.74 ± 0.15 b	30.29 ± 0.22 b
		I	12.35 ± 0.44 B*	4.43 ± 0.24 C*	2.72 ± 0.20 B*	2.29 ± 0.16 C*	18.58 ± 0.50 C*	20.28 ± 0.85 B*
N level (N)			**	**	**	**	**	**
Planting pattern (P)			**	**	**	**	**	**
N × P			**	**	*	**	**	**

The same upper or lowercase letters in the same column indicate significant differences among various nitrogen application levels under intercropping or monocropping planting patterns, respectively, at *p* < 0.05. * indicates that under the same nitrogen fertilizer application rate, the difference between monocropping and intercropping is significant (*p* < 0.05), and ** is extremely significant (*p* < 0.01), and ns is no significant difference. Data are presented as the mean ± standard error of three biological replicates.

**Table 5 plants-12-01232-t005:** Effects of nitrogen application and intercropping on phenolic acid content in faba bean rhizosphere soil.

Site of Measurement	N Level	Planting Pattern (P)	Content of Phenolic Acid (μg/g)					
			PA	VA	FA	BA	SA	CA
Rhizosphere soil	N0	M	11.38 ± 0.17 a	5.85 ± 0.25 a	4.55 ± 0.12 a	4.74 ± 0.11 a	26.48 ± 0.36 a	21.40 ± 0.18 a
		I	9.35 ± 0.15 A*	4.26 ± 0.08 A*	3.82 ± 0.14 A*	3.71 ± 0.21 A*	23.30 ± 0.27 A*	17.61 ± 0.25 A*
	N1	M	9.76 ± 0.08 c	5.37 ± 0.11 b	3.51 ± 0.22 b	3.90 ± 0.23 b	23.79 ± 0.44 c	16.73 ± 0.24 c
		I	6.70 ± 0.20 B*	4.03 ± 0.37 A*	2.23 ± 0.12 B*	2.64 ± 0.23 B*	20.52 ± 0.25 C*	13.92 ± 0.21 C*
	N2	M	8.73 ± 0.14 d	4.06 ± 0.08 d	2.27 ± 0.06 c	2.49 ± 0.08 c	20.58 ± 0.60 d	14.56 ± 0.07 d
		I	5.51 ± 0.34 C*	2.46 ± 0.18 C*	1.25 ± 0.12 C*	1.67 ± 0.10 C*	15.58 ± 0.16 D*	10.97 ± 0.24 D*
	N3	M	10.37 ± 0.09 b	4.83 ± 0.10 c	3.46 ± 0.13 b	3.87 ± 0.11 b	24.88 ± 0.17 b	17.64 ± 0.20 b
		I	9.09 ± 0.16 B*	3.09 ± 0.30 B*	2.47 ± 0.26 B*	2.52 ± 0.19 B*	22.25 ± 0.29 B*	15.98 ± 0.32 B*
N level (N)			**	**	**	**	**	**
Planting pattern (P)			**	**	**	**	**	**
N × P			**	ns	ns	ns	**	**

The same upper or lowercase letters in the same column indicate significant difference among various nitrogen application levels under intercropping or monocropping planting patterns, respectively, at *p* < 0.05. * indicates that under the same nitrogen fertilizer application rate, the difference between monocropping and intercropping is significant (*p* < 0.05), and ** is extremely significant (*p* < 0.01), and ns is no significant difference. Data are presented as the mean ± standard error of three biological replicates.

**Table 6 plants-12-01232-t006:** Effects of nitrogen application and intercropping on the dry weight of the above ground part of the faba bean.

N Level (N)	Planting Pattern (P)	Dry Weight of above Ground Part of the Faba Bean (g)		
		Branching Stage	Flowering Stage	Maturity Stage
N0	M	9.72 ± 0.17 b	11.82 ± 0.85 c	12.65 ± 0.89 d
	I	9.81 ± 0.38 C	18.65 ± 0.88 C*	19.57 ± 1.54 C*
N1	M	10.29 ± 0.20 b	18.63 ± 1.04 b	19.60 ± 0.90 c
	I	10.83 ± 0.42 B	25.69 ± 1.00 B*	27.92 ± 1.49 B*
N2	M	13.16 ± 0.65 a	23.05 ± 1.34 a	24.85 ± 1.12 a
	I	14.12 ± 0.57 A*	27.78 ± 1.01 A*	31.46 ± 0.81 A*
N3	M	12.89 ± 0.82 a	21.85 ± 1.24 a	22.78 ± 1.61 b
	I	13.47 ± 0.65 A	24.15 ± 1.77 B*	27.10 ± 0.63 B*
N level (N)		**	**	**
Planting pattern (P)		*	**	**
N × P		ns	*	ns

The same big (small) and different letters in the same column indicate that there was a significant difference among different nitrogen application levels under intercropping (monocropping) planting patterns at 0.05 level. * means significant difference between monocropping and intercropping at the same nitrogen application rates at 0.05 level. ** means extremely significant difference at 0.01 level and ns is no significant difference. The data are presented as the mean ± standard error of three biological replicates.

## Data Availability

All data generated or analyzed during this study have been included in this published article.

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
