# Peer review of "Nitrogen Fertilization in a Faba Bean–Wheat Intercropping System Can Alleviate the Autotoxic Effects in Faba Bean"

_plants, 2023, doi:10.3390/plants12061232_

Round 1

Reviewer 1 Report

Dear Authors

This is an interesting research problem. It well addressed. The findings would significantly contribute to the current body of science in the specific field in particular and science in general. Please see below my few comments.

Abstract

The methods and results need to be reorganized because they are flawed.
Is it not possible to combine together the methods both experiments and go for the results of both? If not, it is OK and is acceptable in its current form

Authors should also think of combining the two objectives into one (if there is a possibility). Other wise it is OK.

In the abstract, I don't see how the study's goal might be met:
This research offered a theoretical foundation for creating a realistic and highly effective faba bean-wheat intercropping model (Line 97-98)

It could be a good idea to mention that the initial research topic or target was addressed through pot experiments.

I'm not sure if the planned cropping pattern model was created, and nothing was even mentioned in the abstract's results or conclusion.

Author Response

Responses to reviewer1:

_The methods and results need to be reorganized because they are flawed.

Is it not possible to combine together the methods both experiments and go for the results of both? If not, it is OK and is acceptable in its current form.

A: Dear reviewer, thank you very much for your comments. It was our mistake not to explain clearly in the manuscript and we have revised the above problems in the manuscript.

_Authors should also think of combining the two objectives into one (if there is a possibility). Other wise it is OK.

A: Dear reviewer, thank you very much for your comments.

_In the abstract, I don't see how the study's goal might be met:

This research offered a theoretical foundation for creating a realistic and highly effective faba bean-wheat intercropping model (Line 97-98)

A: Dear reviewer, thank you very much for your comments. It was our mistake not to explain clearly in the manuscript and we have revised the above problems in the manuscript.

_It could be a good idea to mention that the initial research topic or target was addressed through pot experiments.

A: Dear reviewer, thank you very much for your comments.

_I'm not sure if the planned cropping pattern model was created, and nothing was even mentioned in the abstract's results or conclusion.

A: Dear reviewer, thank you very much for your comments. It was our mistake not to explain clearly in the manuscript and we have revised the above problems in the manuscript.

Reviewer 2 Report

Review of the article entitled “Nitrogen fertilization in faba bean–wheat intercropping system can alleviate the autotoxic effects in faba bean”

I like the subject of the research and its justification described in the Introduction.

However, I have a lot of comments, especially about the chapters Materials and methods and Results.

Lines 107-109:  What was tested in this pot experiment?

Line 111: Why were Fusarium-infected plants collected for the preparation of the water extract? This should be clearly explained. If Fusarium wilt is so important, a relevant paragraph should be about it in the Introduction chapter

Line 122: About “continuous cropping”? It is necessary to provide information on how many years faba bean had been grown continuously.

Line 187: In this paragraph, the number of plots and their size should be given.

Lines 206-207: Were healthy or diseased plants sampled? What diseases are the authors referring to?

Throughout the Results chapter, I don't like the "The significance reduction" given in % as in the lines 290-297 or 311-338.  Too many numbers in the text discourage a reader. Moreover is it really necessary to give percentages with an accuracy of two decimal places?

When describing the results, the authors should indicate the most important data from the tables or figures.  For me, interesting information is, for example, the data in Table 2:  the content of phenolic acids usually increased with increasing nitrogen dose, but only up to the level of N2. However after the use of N3, the acid contents decreased to the level like after application of N0 or N1. This rule applied to both cropping systems.

In the Results chapter, there is no sentence informing about the general health condition of the plants.

I wonder why there is "intercropping" in the titles of tables 2-6 (what about monocropping?). I think a better term would be "cropping system".

In Conclusion Authors write “The aqueous extract of faba bean roots, stems, leaves, and rhizosphere soil exhibited autotoxic effects on the seed germination”. However, more interesting and important, would be the extract from which part of the plant shows the greatest and which the least autotoxic effect. A similar suggestion applies to the second sentence (lines 528-530).

Other remarks

The article requires many technical corrections, among others:

Key words: do not repeat words that are in the title of the article;

Latin names of species should be written in italics for example Perilla frutescens (line 67), Panax quinquefolium (lines 68-69);

In the Introduction, in many cases the year of publication is not given, e.g. in a line 45 is “Zhou et al. “ but should be “Zhou et al. (2016)”.

In the subtitles, the authors use lowercase and uppercase letters, but it is not known why and according to what rule.

In References very often is written “et al.”. In my opinion all the authors should be mentioned.

Author Response

Responses to reviewer 2:

_Lines 107-109:  What was tested in this pot experiment?

A: Dear reviewer, thank you very much for your comments. It was our mistake not to explain clearly in the manuscript and we have revised the above problems in the manuscript.

_Line 111: Why were Fusarium-infected plants collected for the preparation of the water extract? This should be clearly explained. If Fusarium wilt is so important, a relevant paragraph should be about it in the Introduction chapter.

A: Dear reviewer, thank you very much for your comments. It was our mistake not to explain clearly in the manuscript and we have revised the above problems in the manuscript. We give the following answer to this question. Long-term continuous cropping of faba beans will lead to the occurrence of Fusarium wilt and hinder the growth of faba beans. One of the main reasons for the occurrence of Fusarium wilt is the accumulation of autotoxins. Therefore, the faba bean plants infected with Fusarium wilt accumulated moreautotoxins, so we chose the faba bean plants infected with Fusarium wilt to prepare the water extract.

_Line 122: About “continuous cropping”? It is necessary to provide information on how many years faba bean had been grown continuously.

A: Dear reviewer, thank you very much for your comments. It was our mistake not to explain clearly in the manuscript and we have revised the above problems in the manuscript.

_Line 187: In this paragraph, the number of plots and their size should be given.

A: Dear reviewer, thank you very much for your comments. It was our mistake not to explain clearly in the manuscript and we have revised the above problems in the manuscript.

_Lines 206-207: Were healthy or diseased plants sampled? What diseases are the authors referring to?

A: Dear reviewer, thank you very much for your comments. It was our mistake not to explain clearly in the manuscript and we have revised the above problems in the manuscript.

_Throughout the Results chapter, I don't like the "The significance reduction" given in % as in the lines 290-297 or 311-338.  Too many numbers in the text discourage a reader. Moreover is it really necessary to give percentages with an accuracy of two decimal places?

A: Dear reviewer, thank you very much for your comments. It was our mistake not to explain clearly in the manuscript and we have revised the above problems in the manuscript.

_When describing the results, the authors should indicate the most important data from the tables or figures.  For me, interesting information is, for example, the data in Table 2:  the content of phenolic acids usually increased with increasing nitrogen dose, but only up to the level of N2. However after the use of N3, the acid contents decreased to the level like after application of N0 or N1. This rule applied to both cropping systems.

A: Dear reviewer, thank you very much for your comments. It was our mistake not to explain clearly in the manuscript and we have revised the above problems in the manuscript.

_In the Results chapter, there is no sentence informing about the general health condition of the plants.

A: Dear reviewer, thank you very much for your comments. We give the following answer to this question.In this study, we mainly studied the effects of autotoxins of faba bean on the germination of faba bean seeds, and the effects of nitrogen fertilizer on the content of autotoxins of faba bean and the aboveground biomass of faba bean in the intercropping system. It mainly uses the relevant indicators of seed germination to reflect the inhibition effect of autotoxins on the growth of faba bean, which has no great relationship with the health status of faba bean, so we do not have a sentence about the general health status of plants in the results chapter.

_I wonder why there is "intercropping" in the titles of tables 2-6 (what about monocropping?). I think a better term would be "cropping system".

A: Dear reviewer, thank you very much for your comments. We give the following answer to this question. Long-term monocropping of faba bean will lead to the accumulation of autotoxins, and intercropping is an effective measure to reduce autotoxins. In order to compare the difference in the content of autotoxins between monocropping and intercropping faba beans and reflect the effect of intercropping on reducing autotoxins, so we used “intercropping” in the title of table 2-6, rather than “monocropping” or “cropping system”.

_In Conclusion Authors write “The aqueous extract of faba bean roots, stems, leaves, and rhizosphere soil exhibited autotoxic effects on the seed germination”. However, more interesting and important, would be the extract from which part of the plant shows the greatest and which the least autotoxic effect. A similar suggestion applies to the second sentence (lines 528-530).

A: Dear reviewer, thank you very much for your comments. It was our mistake not to explain clearly in the manuscript and we have revised the above problems in the manuscript.

_Key words: do not repeat words that are in the title of the article

A: Dear reviewer, thank you very much for your comments. It was our mistake not to explain clearly in the manuscript and we have revised the above problems in the manuscript.

_Latin names of species should be written in italics for example Perilla frutescens (line 67), Panax quinquefolium (lines 68-69);

A: Dear reviewer, thank you very much for your comments. It was our mistake not to explain clearly in the manuscript and we have revised the above problems in the manuscript.

_In the Introduction, in many cases the year of publication is not given, e.g. in a line 45 is “Zhou et al. “ but should be “Zhou et al. (2016)”.

A: Dear reviewer, thank you very much for your comments. The citation format of the references in the introduction is according to the requirements of the journal.

_In the subtitles, the authors use lowercase and uppercase letters, but it is not known why and according to what rule.

A: Dear reviewer, thank you very much for your comments. It was our mistake not to explain clearly in the manuscript.We give the following answer to this question. The data in this part comes from the field experiment, and our field experiment adopts the two-factor design. In order to distinguish the significant differences between different factors, we use capital letters and lowercase letters to express the significant differences between different factors.In the subtitles,the same capital letters in the same column of the table indicate the significant differences between different nitrogen application levels under intercropping mode; ,the same lowercase letters in the same column indicate the significant difference between different nitrogen fertilizer application levels under monocropping mode (p<0.05); * it indicates that under the same nitrogen fertilizer application rate, the difference between monocropping and intercropping is significant (p<0.05), and * * is extremely significant (p<0.01). No marked with * indicates no significant difference.

_In References very often is written “et al.”. In my opinion all the authors should be mentioned.

A: Dear reviewer, thank you very much for your comments. The citation format of the references in the introduction is according to the requirements of the journal.

Reviewer 3 Report

There are serious mistakes in the manuscript, and major revisions are required to meet review standards. For example, in Table 6, this article does not involve disease research, and it is not mentioned in Materials and Methods, but in Table 6, different stages of disease appear. The discussion on lines 492 to 496 is not relevant to the article. Please revise and resubmit the MS carefully.

How to prove that the toxins by the water extracts are related to the amount contained in the plants themselves? Instead of using other chemicals to extract?

Phenolic acids certainly have a limiting effect on seed germination at high concentrations, but it is difficult for plants to produce such high concentrations of phenolic acids. How to explain this autotoxicity comes from phenolic acids?

Why didn't it prove that phenolic acids have a limiting effect on crop growth? And only did the germination of the seeds?

The format of the figure needs to be modified according to the requirements of the journal.

Extensive editing of English language and style required.

Author Response

Responses to reviewer 3:

_There are serious mistakes in the manuscript, and major revisions are required to meet review standards. For example, in Table 6, this article does not involve disease research, and it is not mentioned in Materials and Methods, but in Table 6, different stages of disease appear. The discussion on lines 492 to 496 is not relevant to the article. Please revise and resubmit the MS carefully.

A: Dear reviewer, thank you very much for your comments. It was our mistake not to explain clearly in the manuscript. In Table 6, the disease stage of emergence is actually the time we sampled and does not involve the study of disease, so there is no investigation of disease present in the material method. The three disease stages in Table 6 correspond to different growth stages of faba bean, respectively, the branching stage, flowering stage, and maturity stage.We have revised the above problems in the manuscript. In addition, regarding the contents of lines 492 to 496, we give the following explanation. In our results, we found that intercropping of wheat and faba bean can significantly reduce the content of phenolic acid in faba bean, but there are few studies on the mechanism of intercropping to reduce the content of phenolic acid. This is also the part we need to think about. When consulting the literature, we found that the reduction of phenolic acid content by intercropping might be related to soil microorganisms. Therefore, we speculate that the reason why wheat and broad bean intercropping can significantly reduce broad bean phenolic acid is also related to soil microorganisms. This may be the future research direction, so we mentioned this part in the discussion.

_How to prove that the toxins by the water extracts are related to the amount contained in the plants themselves? Instead of using other chemicals to extract?

A: Dear reviewer, thank you very much for your comments. We give the following answer to this question. Because compared with other organic solvent extracts, the water extracts of plants can directly reflect the autotoxicity of plants (Yu et al. 2000). Therefore, we selected the water extract of faba bean plant and rhizosphere soil to test the autotoxicity of faba bean.When we prepare the water extract, we directly extract the autotoxins from the faba bean plant and the rhizosphere soil with water. There is no addition or destruction of other chemical reagents in the preparation process, so the self-toxic substances in the water extract are from the plant itself.

(Yu J Q, Shou S Y, Qian Y R, et al. Autotoxic potential of cucurbit crops. Plant and Soil, 2000, 223(1): 149-153.)

_Phenolic acids certainly have a limiting effect on seed germination at high concentrations, but it is difficult for plants to produce such high concentrations of phenolic acids. How to explain this autotoxicity comes from phenolic acids?

A: Dear reviewer, thank you very much for your comments. We give the following answer to this question. Because in the 6-year field experiment, we found that the content of phenolic acid increased with the increase of continuous cropping years. The content of salicylic acid in the sixth year reached 20μg.g-1, which is only the result of continuous planting for six years. With the increase of continuous planting, the content of phenolic acid will be higher. There are many soil microorganisms in the field soil, which use phenolic acid as food, and the consumption of soil microorganisms is very large (Liu et al. 2017). So we think that the content of phenolic acid in plant rhizosphere should be much higher than our measured value. So we designed phenolic acid gradients of 50, 100, 200, 400 and 800 mg · L-1.

(Liu,J.,Li,X.,Jia,Z.,Zhang,T.,&Wang,X..(2016).Effectofbenzoicacidonsoilmicrobialcommunitiesassociatedwithsoilbornepeanutdiseases.AppliedSoilEcology,110(Complete),34-42.)

_Why didn't it prove that phenolic acids have a limiting effect on crop growth? And only did the germination of the seeds?

A: Dear reviewer, thank you very much for your comments. We give the following answer to this question. Because seed germination is an important prerequisite for plant growth, it regulates the yield and productivity of plants and is of great significance to the growth, development and yield of later plants. The results are representative. Germination energy is the most direct indicator of seed germination ability; Germination rate can reflect the growth ability of seeds; Germination index can reflect the resistance of seeds to adversity and is an important indicator to guide crop growth. So we only measured the effect of phenolic acid on seed germination.Crop growth and seed germination are similar in a sense, and our research results show that phenolic acid has a restrictive effect on faba bean seed germination, which can explain our scientific problems. In addition, we currently have no data on crop growth indicators.

_The format of the figure needs to be modified according to the requirements of the journal.

A: Dear reviewer, thank you very much for your comments. It was our mistake not to explain clearly in the manuscript and we have revised the above problems in the manuscript.

_Extensive editing of English language and style required.

A: Dear reviewer, thank you very much for your comments. It was our mistake not to explain clearly in the manuscript and we have revised the above problems in the manuscript.

Round 2

Reviewer 2 Report

The authors followed my comments.

I still think the description of the results is not perfect, but it is acceptable.

Reviewer 3 Report

Accept in present form.